# Upgrading Monocytes Therapy for Critical Limb Ischemia Patient Treatment: Pre-Clinical and GMP-Validation Aspects

**DOI:** 10.3390/ijms232012669

**Published:** 2022-10-21

**Authors:** Giulio Rusconi, Giuseppe Cusumano, Luca Mariotta, Reto Canevascini, Mauro Gola, Rosalba Gornati, Gianni Soldati

**Affiliations:** 1Swiss Stem Cell Foundation, 6900 Lugano, Switzerland; 2Department of Surgery, Service of Angiology, Lugano Regional Hospital, 6900 Lugano, Switzerland; 3Department of Biotechnology and Life Sciences, University of Insubria, 21100 Varese, Italy

**Keywords:** PBMC, critical limb ischemia, ATMP, cell therapy, monocyte, GMP production, Ficoll

## Abstract

Advanced cell therapy medicinal products (ATMP) are at the forefront of a new range of biopharmaceuticals. The use of ATMP has evolved and increased in the last decades, representing a new approach to treating diseases that are not effectively managed with conventional treatments. The standard worldwide recognized for drug production is the Good Manufacturing Practices (GMP), widely used in the pharma production of synthesized drugs but applying also to ATMP. GMP guidelines are worldwide recognized standards to manufacture medicinal products to guarantee high quality, safety, and efficacy. In this report, we describe the pre-clinical and the GMP upgrade of peripheral blood mononuclear cell (PBMC) preparation, starting from peripheral blood and ending up with a GMP-grade clinical product ready to be used in patients with critical limb ischemia (CLI). We also evaluated production in hypoxic conditions to increase PBMC functional activity and angiogenic potential. Furthermore, we extensively analyzed the storage and transport conditions of the final product as required by the regulatory body for ATMPs. Altogether, results suggest that the whole manufacturing process can be performed for clinical application. Peripheral blood collected by a physician should be transported at room temperature, and PBMCs should be isolated in a clean room within 8 h of venipuncture. Frozen cells can be stored in nitrogen vapors and thawed for up to 12 months. PBMCs resuspended in 5% human albumin solution should be stored and transported at 4 °C before injection in patients within 24 h to thawing. Hypoxic conditioning of PBMCs should be implemented for clinical application, as it showed a significant enhancement of PBMC functional activity, in particular with increased adhesion, migration, and oxidative stress resistance. We demonstrated the feasibility and the quality of a GMP-enriched suspension of monocytes as an ATMP, tested in a clean room facility for all aspects related to production in respect of all the GMP criteria that allow its use as an ATMP. We think that these results could ease the way to the clinical application of ATMPs.

## 1. Introduction

The use of advanced therapy medicinal products (ATMP) has evolved and increased in the last decades, representing a new approach to treating diseases that are not effectively managed with conventional treatments. ATMP cannot be subjected to regulations of conventional and chemically synthesized drugs, as their manufacturing can be complex and potentially subjected to contamination [1]. GMP guidelines are worldwide recognized standards to manufacture medicinal products to guarantee high quality, safety, and efficacy. They oversee the entire life cycle of a product, from manufacturing to distribution and supply, as they are necessary to obtain marketing authorization [2]. Clean rooms are suitable facilities for producing ATMPs following GMP guidelines. Validations are planned experiments necessary to prove control of all critical aspects, processes, materials, and equipment related to intermediate and finished products. They require a document defining all necessary experiments and acceptance criteria for validation (validation master plan), as well as validation reports summarizing the results [3].

Peripheral blood mononuclear cells (PBMC) are mainly represented by lymphocytes and monocytes, with an inter-individual variation of frequencies, which are usually in the range of 70% to 90% for lymphocytes and 10% to 20% for monocytes. Additional and less abundant cell types are also classified as PBMC, such as dendritic cells (DC), endothelial progenitor cells (EPC), and hematopoietic stem cells (HSC), that are found in low percentages in blood [4,5]. Monocytes are recruited into tissues by inflammatory signals following the model of leukocyte adhesion and trafficking: rolling, adhesion, and transmigration [6]. Macrophages are differentiated and tissue-resident cells, acting as multifunctional phagocytes in various physiological and pathological conditions. The most important functions of macrophages are immune surveillance, tissue homeostasis, reaction to infection, induction, and resolution of inflammation, as well as wound healing [7]. They are strictly related to the tissue micro-environment and external stimuli, inducing differentiation and specialization of phenotypes and functions [8]. Macrophages are responsible for the secretion of numerous cytokines, chemokines, metalloproteinases, and growth factors inducing inflammation, tissue repair, and fibrosis [9], as well as angiogenesis and arteriogenesis [10]. VEGF-A is the most relevant cytokine involved in new blood vessel formation in adults, acting through VEGFR 1 and 2 expressed on endothelial cells [11]. In vitro studies showed that VEGF promotes proliferation, migration, mitochondrial function, and angiogenic potential of endothelial cells [12]. Furthermore, VEGF and its role in neovascularization are strictly related to ischemic tissues, where angiogenesis needs to be stimulated. Indeed, hypoxia-induced VEGF production is a fundamental mechanism occurring in physiological and pathological conditions of low oxygen concentrations [13].

Peripheral arterial disease (PAD) is a pathological condition characterized by insufficient blood and oxygen supply in the lower limbs, which can cause tissue damage and dysfunction. It is estimated that 200 million people worldwide are affected by PAD, and the incidence increases to 20% of people over the age of 70 and 40% of people over the age of 80 [14]. Critical limb ischemia (CLI) is the most severe form of PAD, characterized by chronic pain, ulceration, and gangrene, finally leading to limb amputation and death. CLI has an incidence of 500 to 1000 cases per million individuals and is usually related to a bad prognosis in terms of limb preservation and survival rate [15]. Recent development and progress in surgical techniques have improved limb salvage and survival rates of patients affected by CLI. Nonetheless, a significant percentage of patients called no-option are subjected to amputation and suffer from poor quality of life [15]. High morbidity and mortality of CLI generate interest in innovative therapeutic approaches for restoration of blood flow and tissue repair. The mechanism of monocytes and macrophages to promote vascular remodeling has been investigated for decades. Several studies reported the association of monocytes and macrophages with the collateral growth of blood vessels in animal models. In particular, accumulations were observed in growing arteries in ischemic hindlimbs in rabbits [16] and mice [17]. In addition, the role of blood-derived monocytes as endothelial progenitors has been described in vitro in different studies [18,19]. The adhesion ability of monocytes to the endothelium was associated with collateral arterial growth and angiogenesis, as they are recruited to the endothelium by ischemic or inflammatory signals. Adhesion and migration into the subendothelial space allow for monocyte maturation to macrophages and subsequent production of growth factors and cytokines, including MCP-1 [10,20]. Apart from angiogenesis, resolution of inflammation and tissue repair are crucial functions of activated macrophages, particularly important for the neovascularization of lesioned limbs [21]. An important advantage of PBMCs is the non-invasive approach of blood collection, compared to risky and invasive bone marrow harvesting, making PBMCs a suitable candidate for cell-based therapy of CLI. Monocytes and macrophages are involved in neovascularization as they accumulate in ischemic and hypoxic regions, where they are stimulated by low oxygen concentrations and the HIF-1 transcription factor [22]. The effect of hypoxia on monocytes has been investigated, showing their ability to adapt to hypoxic environments and increase their angiogenic potential [23]. Hypoxic conditioning in vitro and implantation of PBMCs was successfully performed in animal models of hindlimb ischemia. Hypoxic conditioning of mouse PBMCs enhanced oxidative stress resistance, resulting in increased angiogenic potential and survival rate after injection into ischemic limbs [24]. Furthermore, hypoxia induced the expression of adhesion molecules such as CXCR4 and Integrin αM, increasing the adhesion and retention of PBMCs in ischemic tissues [25]. Similarly, conditioned PBMCs from rabbits resulted in increased adhesion, VEGF production, and oxidative stress resistance in vitro, as well as accelerated neovascularization in ischemic limbs [26]. Furthermore, the conditioning protocol was tested with human PBMC, which showed consistent results in vitro and restored blood flow after implantation in mouse models of hindlimb ischemia [27].

## 2. Results

The experimental design of the study is shown in Appendix A.

### 2.1. Pre-Clinical Experiments

#### 2.1.1. Characterization of PBMC by Flow Cytometry Immunophenotyping

After purification with Ficoll, PBMC phenotyping was performed on the total count of monocytes (CD14), T cells (CD3), B cells (CD19), and NK cells (CD56), as shown in Table 1. Two additional antibodies were used for platelets (CD41) and to assess the absence of granulocytes (CD66b). The gating strategy is illustrated in Appendix A.

#### 2.1.2. Functional Analysis of Cryopreserved PBMCs under Hypoxic Conditions

Several published studies assessed the improved functionality of PBMCs in cells previously conditioned in a hypoxic environment (24–27). Therefore, a 24 h hypoxic conditioning (5% CO_2_, 2% O_2_, 93% N_2_) was performed to increase PBMC functionality. Control samples were cultured in a normoxic environment (5% CO_2_, 95% air) for 24 h. For pre-clinical analysis, a low-cost home-made system for hypoxic conditioning was designed and tested, and the effects of hypoxic conditioning were evaluated by visual microscope inspection, cell count, viability, and immunophenotyping of cells cultured in both conditions. Visual inspection of PBMCs at 24 h under hypoxic conditions did not point out any difference with respect to normoxic conditions, as shown in Figure 1.

Comparing cell number and viability of PBMCs before conditioning and after a 24 h culture in normoxic or hypoxic environments showed that relative cell count in both conditions was slightly lower compared to the number of cells counted before culture (0.70 and 0.69, respectively, Figure 2A); however, no significant difference was observed (*p* = 0.26). Moreover, cell loss was equivalent in samples conditioned in normoxia and hypoxia (*p* = 0.99). The viability percentage of PBMCs resulted lower after 24 h of culture in both conditions; however, no difference was observed between normoxic and hypoxic cultures (*p* = 0.93, Figure 2B). Relative monocyte count was significantly decreased after conditioning (*p* = 0.0008 and *p* = 0.003 compared to Before, Figure 2C) and similar in both conditions (0.73 and 0.78, *p* = 0.90). T-lymphocyte count resulted slightly higher after culture in normoxia and decreased in hypoxic samples (0.78 and 0.66, Figure 2D), compared to cell count before culture (*p* = 0.09 and *p* = 0.009 in normoxia and hypoxia, respectively). Similar outcomes characterized relative B cell counts after conditioning (0.71 and 0.64, respectively, in normoxia and hypoxia, Figure 2E), with a significant decrease in hypoxic samples (*p* = 0.1 and *p* = 0.03 in normoxia and hypoxia, respectively). NK-lymphocytes were significantly decreased by more than 50% after culture in both conditions (*p* = 0.0001 compared to Before, Figure 2F); however, no difference resulted between normoxia and hypoxia (0.46 and 0.43, *p* = 0.90). However, more importantly, the average granulocyte number after conditioning was decreased by 90% compared to freshly thawed samples (*p* = 0.0001, Figure 2G) and similar in both conditions (0.08 and 0.10, *p* = 0.42). Similarly, platelet count resulted significantly lower after culture (*p* = 0.0001, Figure 2H) and comparable in normoxia and hypoxia (0.45 and 0.43, *p*-value = 0.95). Notably, cell count and immunophenotyping of conditioned samples showed similar results in normoxic and hypoxic cultures, as shown in Figure 2.

In the second series of experiments, we evaluated the adhesion and migration potential of conditioned PBMCs with fibronectin-coated plates and 8 µm pore cell culture inserts. Three replicates of 5 × 10^5^ conditioned cells were seeded in fibronectin-coated 48-well plates and incubated for 4 h under normoxic conditions. Non-adherent cells were washed and collected for counting to calculate the percentage of adherent cells. Adherent cells were subsequently fixed, stained with crystal violet, and quantified by colorimetric assay. Both methods showed comparable results, with a slight overestimation of adherent cells with the subtraction approach. Average cell adhesion of PBMCs conditioned in hypoxia was significantly higher compared to control samples cultured in normoxic conditions, as shown in Figure 3. In hypoxic cultures, a percentage of 37 and 35% of adherent cells were calculated by subtraction and crystal violet staining, respectively. Adhesion percentage calculated in normoxic controls with both methods was significantly lower, 22 and 23% respectively (*p* = 0.02 and *p* = 0.009, Figure 3A,B). Three replicates of 3 × 10^5^ conditioned cells seeded on polycarbonate cell culture inserts were exposed to MCP-1 as a chemoattractant, and migration through a membrane was evaluated by counting cells. Average counts showed significant differences in samples conditioned in hypoxia compared to normoxic controls. Figure 3C shows the percentage of migrated cells that resulted higher in hypoxic cultures compared to controls (45 and 37%, respectively, *p* = 0.0006).

To quantify the oxidative stress resistance by PBMCs under hypoxic conditions, we plated triplicates of conditioned cells in 96-well plates at a density of 1 × 10^6^/mL and induced oxidative stress by exposing cells to a solution of 400 µM hydrogen peroxide for 24 h. Viability of hypoxic cultures was significantly higher after treatment (74% and 82%, *p* = 0.0001, Figure 3D). Furthermore, the production of ROS by PBMCs in N = 8 samples was measured after conditioning in normoxic and hypoxic cultures and compared to ROS production after oxidative stress. No significant difference due to conditioning was detected between hypoxic and normoxic samples. Moreover, a 1.6-fold increase in ROS production was induced by oxidative stress in normoxic controls (*p* = 0.048, Figure 3E), whereas no increase was observed in hypoxic samples after treatment. Statistical analysis pointed out a significant difference in ROS production in normoxic and hypoxic samples after oxidative stress (*p* = 0.002).

We finally assessed the expression of genes involved in angiogenesis, inflammation, and adhesion in conditioned PBMCs by quantitative RT-PCR. The most important enhancement was observed in the expression of VEGF, with a 6-fold increase in cells conditioned in hypoxia, compared to normoxic controls (*p* = 0.007, Figure 4A). However, no significant upregulation was detected in mRNA levels of the other growth factors considered, FGF-2 and PDGF, which showed only a slight increase in hypoxic cultures (1.49 and 1.88, *p* = 0.14 and *p* = 0.13, Figure 4B and C, respectively). The expression of both genes encoding for the adhesion molecules CCR2 and CXCR4 were slightly increased after hypoxic conditioning, compared to normoxic control (1.34 and 1.29, Figure 4D,E). Nonetheless, the statistical analysis pointed out a significant difference only in CXCR4 expression (*p* = 0.011) due to a lower variability among replicate experiments. Of the inflammatory cytokines considered in the analysis, TNF-ɑ was the only gene showing similar levels of mRNA in normoxic and hypoxic cultures (*p* = 0.09, Figure 4F). Statistical analysis of IFNG expression levels was strongly significant (*p* = 0.0003, Figure 4G), with low variability among replicates; nonetheless, only a 1.6-fold average increase in hypoxic conditions was observed. Both IL-1b and IL-2 were upregulated at the mRNA level in PBMCs cultured in hypoxia, with more than a 2-fold change compared to controls (*p* = 0.021 and *p* = 0.022, Figure 4H,I).

### 2.2. GMP-Compliant Validation

#### 2.2.1. Validation of Pre-Processing Time and Temperature

The evaluation of cell count and viability is shown in Figure 5, where no significant difference could be observed in cell number and viability after an 8 h storage at room temperature (23 ± 5 °C). The average cell number per ml of blood was decreased by less than 10% after 8 h compared to controls, and both cell count and viability parameters were found to be above the average calculated on 11 pre-clinical samples. Concerning monocyte and granulocyte counts, validation data showed no significant difference in counts after an 8 h storage at room temperature. Average monocyte and granulocyte numbers per ml of blood were found to be above the average calculated on seven pre-clinical samples. Furthermore, the granulocyte percentage was found to be below 1%.

#### 2.2.2. GMP Upgrade of PBMC Isolation, Cryopreservation, and Thawing

Clean room validations aim to verify the possibility of transferring the method for production in an aseptic environment following GMP guidelines, as well as the consistency of results obtained in the pre-clinical process. The validation data shown in Figure 6 were in line with the expectations; the cell number per ml of blood and viability were above the average calculated on 11 different research samples. Monocyte number per ml of blood was above the average calculated on seven different research samples, whereas granulocyte number was below the average.

Validation data in Figure 7 showed an average cell recovery above 80% and similar to the average calculated on 27 thawing processes of seven different research samples. The average viability difference was below 1%. Monocyte recovery was above 80% and slightly lower than the average obtained in the research process. The granulocyte recovery percentage was significantly lower compared to the research.

#### 2.2.3. Validation of Storage Time for Thawed PBMCs before Injection

Validation data shown in Figure 8 did not point out any difference in cell number and viability measured with Nucleocounter. Monocyte count determined by flow cytometry presented no difference as well, whereas granulocyte count was significantly lower after 24 h storage at 4 ± 2 °C. Altogether, the results confirmed that injectable PBMCs could be stored at 4 ± 2 °C for up to 24 h before injection.

#### 2.2.4. Validation of Long-Term Storage of Cryopreserved PBMC

After long-term storage in vapors of liquid nitrogen, cells were thawed and evaluated for cell recovery and viability. Validation data showed cell recovery of at least 75% at each time point as shown in Figure 9. Viability of thawed PBMCs was observed to be higher than 99% in all tested samples (N = 3) at any time point. Monocyte recovery was at least 80% in all thawed samples at each time point. Granulocyte count showed a significant decrease in thawed samples, compared to freshly isolated ones, representing anyway a percentage of cells below 2% with very low absolute numbers.

## 3. Discussion

The last decades have been characterized by a huge revolution in biomedical and biotechnological fields. Biopharmaceutical products represent a new approach for diseases that cannot be effectively treated with conventional treatments, and advanced therapy medicinal products (ATMP) rely on the metabolic effect of the cells implanted to stimulate the self-healing ability of the human body to achieve a therapeutic effect [28]. Critical limb ischemia (CLI) is the most severe form of peripheral arterial disease (PAD), characterized by rest pain, tissue loss, high risk of limb amputation or death, and generally reduced quality of life. The conventional treatments, including drug therapy and vascular surgery, are often not sufficient to avoid complications and consequent amputation of the limb. Furthermore, many patients are not eligible for revascularization, creating a great limitation in treatment [29]. The increasing incidence and mortality of PAD drive scientific research toward new alternative therapies, where autologous BM-MNC and PBMCs seem to be promising sources for the induction of neovascularization in patients affected by cardiovascular diseases [30,31]. However, considering a clinical application in patients, peripheral blood represents a valuable alternative source, as it is more accessible and collected with a safe and minimally-invasive procedure [32]. Moreover, the role of monocytes and macrophages in angiogenesis and tissue repair is a therapeutic approach worth considering for the treatment of CLI.

### 3.1. Pre-Clinical Experiments

PBMC isolation from peripheral blood was performed by density gradient centrifugation with Ficoll^®^-Paque PREMIUM, according to reported methods and studies [33]. Different systems were described to obtain PBMCs from peripheral blood in the operating room, in particular, apheresis with cell separators [34] and selective filtration [35]. These methods are characterized by variable contaminations of granulocytes, which are not always separated from mononuclear cells [36]. An important feature of PBMCs isolated with Ficoll^®^ is the exclusion of granulocytes, particularly neutrophils [37]. Indeed, it has been reported that neutrophils can interact with monocytes and prevent their differentiation into dendritic cells, reprogramming them into anti-inflammatory macrophages. Moreover, neutrophil extracellular traps rendered monocytes less efficient in killing pathogens [38]. The advantage of density gradient centrifugation with Ficoll^®^ is the efficiency in terms of selectivity and purity of isolated mononuclear cells, with a certified and GMP-compliant material to manufacture the product for clinical use.

Characterization of the PBMC product was performed by flow cytometry immunophenotyping. The average proportion and absolute count of all PBMC populations were compared before and after cryopreservation and thawing. Monocyte percentage was increased in thawed samples, whereas the other cell types showed similar proportions before freezing and after thawing. The average cell count per ml of processed blood showed a 30% loss of T lymphocytes, platelets, and granulocytes during cryopreservation. With respect to PBMCs, flow cytometry has been previously performed, either in whole blood [39,40] or in samples isolated by density gradient centrifugation [41], where the vast majority of monocytes express CD14 [42]. We also selected three antibodies to stain lymphocytes: CD3 for T cells, CD19 for B cells, and CD56 for NK cells [41]. In addition, CD41 was selected for platelets [43] and CD66b for granulocytes [44].

#### Functional Analysis of Cryopreserved PBMCs under Hypoxic Conditions

Hypoxic conditioning was described as a potent stimulus of the angiogenic potential of PBMCs, and functional activity is thus an important feature worth considering when using PBMCs for therapeutic neovascularization [24]. Cryopreserved PBMCs were exposed to a low oxygen concentration, and visual observation after 24 h of culture in normoxic or hypoxic conditions did not show any difference in cell morphology and cell density. However, we found a slight decrease in the number of nucleated cells and viability down to 90%, which is anyway above the level of 70% fixed by the European Pharmacopoeia. Monocyte count resulted in a decrease of more than 20% in both normoxic and hypoxic conditions, in line with the lower nucleated cell count observed. Average T and B cell counts resulted slightly lower in hypoxic cultures compared to normoxic controls and significantly decreased compared to cell count before culture. Furthermore, NK cells were reduced by an average of 50% after culture in both conditions. Among the lymphocyte subpopulation, NK seemed to be the more affected by a 24 h culture, whereas oxygen concentration did not induce any changes in CD56 expression. Average granulocyte count pointed out a cell loss of 90% in normoxic and hypoxic conditions, compared to cell count before culture. Indeed, the role of neutrophils in innate immunity and inflammation is related to programmed cell death, which is required to limit neutrophil action. Moreover, the recognition of apoptotic cells by macrophages is essential for the resolution of inflammation [45]. Average platelet counts after culture were reduced by 50% in both conditions. Nonetheless, platelets can be considered as a supplement in our product; therefore, a decreased number did not raise concerns about quality and effectiveness. The similarity of results obtained in both conditions did not suggest a detrimental effect of hypoxia on PBMC number, viability, phenotype, and monocyte count.

Adhesion and migration potentials of PBMCs are crucial functions related to therapeutic neovascularization. Indeed, monocytes are known to be recruited and attach to the endothelium before they migrate into the subendothelial layer and differentiate into macrophages [20]. The adhesion capacity of conditioned PBMCs was evaluated by incubation on fibronectin-coated plates in normoxic conditions. Non-adherent cells were washed after 4 h of incubation and counted, whereas adherent cells were fixed and stained for further quantification. Both methods showed consistent results, pointing out a significant increase in adherent cells in hypoxic cultures, compared to normoxic controls, as also previously reported, demonstrating increased adhesion capacity of mouse [25], rabbit [26], and human PBMCs in vitro [27]. Migration potential was assessed by chemotaxis with hPL and MCP-1. We observed a significant difference in migrated and non-migrated cells between normoxic and hypoxic cultures, therefore suggesting that active migration towards MCP-1 could be enhanced in hypoxia-conditioned PBMC.

Resistance to oxidative stress is a fundamental feature of cells implanted in ischemic limbs, as they are usually subjected to a hypoxic environment. Therefore, successful angiogenic induction depends on the survival of implanted cells exposed to oxidative stress [24]. Our results showed that hypoxic conditioning increased cell viability after treatment with hydrogen peroxide, compared to normoxic controls. Furthermore, we evaluated ROS production in PBMCs after conditioning in normoxia or hypoxia and after oxidative stress induced by hydrogen peroxide. Our results showed similar levels of intracellular ROS after normoxic and hypoxic conditioning, suggesting adaptation of PBMCs to the hypoxic environment. After 24 h of oxidative stress, normoxic controls showed an increased ROS production that was not evident in hypoxic cells. Several studies investigated PBMCs and BM-MNC resistance to oxidative stress in vitro, using animal models [24,26,46] and human cells [27] but not in hypoxic conditions. Reduced ROS accumulation was proposed as a possible explanation for the increased PBMC survival rate [24,26]. Indeed, the role of ROS as a mediator of several cellular functions has been extensively described. The main processes are cell growth, proliferation, differentiation, and apoptosis, and it is generally acknowledged that increased ROS is associated with cell death [47,48]. Moreover, ROS has been associated with tissue injury and several acute and chronic diseases, including cardiovascular pathologies [49]. Interestingly, mitochondrial production of ROS has been shown to regulate the response of cells to hypoxia [50].

Hypoxic conditioning has been shown to upregulate the expression of genes related to cell survival and antioxidants [24]. We studied mRNA expression of genes involved in angiogenesis, inflammation, and adhesion of PBMCs in conditioned cells. We evaluated VEGF, FGF-2, and PDGF, which are among the most important angiogenic growth factors regulated by the HIF-1 system. VEGF was the only gene strongly upregulated in hypoxic cultures, with a 6-fold increase, whereas FGF-2 and PDGF showed similar levels in both conditions. Hypoxia-induced VEGF production was extensively reviewed in many publications and described as one of the most important mechanisms of neovascularization in physiologic and pathologic conditions, in particular in tumors [11,51]. Moreover, enhanced VEGF expression was reported after a 24 h hypoxic conditioning in mouse BM-MNC [46], as well as PBMCs of mouse [24] and human origin [26]. We further analyzed genes coding for adhesion molecules, and CCR2 and CXCR4 mRNA were quantified in both normoxic and hypoxic conditions. We found both genes slightly upregulated in hypoxic cultures (1.3-fold) with CXCR4 significantly increased. CCR2 encodes for the MCP-1 chemokine receptor and is expressed by monocytes, with a fundamental role in their recruitment to inflammatory sites [52]. CXCR4 and its ligand SDF-1 regulate lymphocyte homeostasis and activation, as well as migration toward inflammatory sites and homing [53]. Moreover, SDF-1-dependent migration has been reported in monocytes as well [54]. Interestingly, the evaluation of CXCR4 mRNA expression showed a HIF-1-dependent upregulation in response to hypoxia [55]. Moreover, a 24 h hypoxic conditioning of PBMCs increased CXCR4 expression, as well as cell retention and angiogenic induction in ischemic limbs after implantation in mice [25]. A similar outcome was observed in rabbit PBMCs exposed to a 24 h hypoxic conditioning [26], whereas a 2-fold induction was reported in mice with a similar protocol [25]. Regulation of inflammation is crucial for neovascularization; therefore, we measured the mRNA expression of inflammatory cytokines to evaluate the inflammatory potential of conditioned cells. TNF-α was the only gene showing similar mRNA levels in both normoxic and hypoxic conditions, whereas IFN-γ, IL-1β, and IL-2 were significantly upregulated. Our experiments presented very low variability in IFN-γ expressions among replicates; therefore, a 1.5-fold induction was corroborated by strong statistical analysis. We found a 2-fold induction in gene expression of both IL-1β and IL-2 under hypoxic conditions, suggesting a possible activation of PBMCs and the HIF-1 transcription factor. Our analysis gave some insight into the effect of hypoxic conditioning on the expression of genes related to angiogenesis, cell adhesion, and inflammation. Only VEGF showed a strong 6-fold enhancement induced by hypoxia. Indeed, VEGF is the most relevant growth factor in neovascularization, and HIF-1-dependent induction of VEGF has been extensively described [56]. Other mRNA upregulations were moderate, leaving some doubts about a possible therapeutic improvement mediated by increased adhesion molecules or inflammatory cytokines.

### 3.2. GMP-Compliant Validation

Validations were planned to demonstrate the similarity of outcomes in different situations or time points. Blood storage during transportation was identified as a key parameter affecting the quality of samples. Blood collection, storage, and transport are fundamental parameters affecting the quality of the finished product, especially when used as raw material for cell isolation. In order to guarantee laboratory operativity, it is, however, important to identify the optimal time frame. Decreased cell recovery, viability, and functional activity of PBMCs isolated 24 h after venipuncture, compared to cells isolated within 8 h, was already reported [57]. Granulocyte contamination and decreased T cell function were observed when blood stored at room temperature was processed with an 18 h delay [58]. Granulocyte contamination was shown to be increased over time at room temperature and correlated with decreased T cell proliferation. Furthermore, a significant increase in granulocyte count and T cell inhibition was pointed out after 24 h but not within 8 h of blood collection [59]. More recent studies indicated that 12 h at 15 °C and 40 °C were sufficient to decrease cell yield, viability, and function, compared to samples stored at 22 °C and 30 °C [60]. Therefore, storage time and temperature were established according to previous studies, showing consistent results within 8 h of collection, as previously described [57,58,59,60,61]. Validation was performed by splitting three blood samples and processing them immediately or after 8 h at room temperature (23 ± 5 °C). Results were encouraging, and a maximum of 20% decrease in cell count was measured in a sample. Nonetheless, viability and granulocyte contamination were similar, with granulocyte numbers below 1%. Average values demonstrated that peripheral blood could be transported for up to 8 h at room temperature before processing in a cell factory.

Validation of aseptic manufacturing compliant with GMP guidelines was performed in an authorized clean room facility for both isolations of PBMCs and for the cryopreservation process, where thawed cells were further analyzed. Intermediate and finished products were assessed for aseptic production and sterility, along with environmental and microbiological monitoring. Aseptic production was confirmed by environmental and microbiological controls, and no growth of microorganisms was detected in any of the tested samples. Fresh and thawed PBMCs were investigated for nucleated cell count and viability, as well as monocyte and granulocyte count, by immunophenotyping. All measured parameters met the expectations, showing comparable or better outcomes compared to the pre-clinical research process, demonstrating that the whole process is suitable for GMP-compliant manufacturing.

The conditions of transporting back to the patient’s bed were tested for 24 h at 4 °C, considering that a maximum transport time of 24 h would be a technical advantage. The resuspension medium of 5% Human Albumin solution was found to be suitable in terms of GMP guidelines, as human albumin is well accepted in the formulation of the final product and because it represents a good transport medium for cells. We observed, indeed, cell count, viability, and monocyte count were similar immediately after thawing and 24 h later, whereas granulocytes were significantly decreased after storage. Indeed, neutrophil apoptosis could explain the reduction, as previously hypothesized [45]. Nonetheless, a decreased contamination of granulocytes could be considered a positive outcome.

Since sterility tests of the final product are based on microbiological cultures, which take 14 days to be completed before the final ATMP product can be released for the patient, we suggest cryopreserving PBMCs before reinfusion, providing thus a valuable time to obtain results and deliver a fully characterized ATMP product. Another advantage of cryopreservation is the possibility of multiple thawing and repeated injections over time in order to increase therapeutic neovascularization. Blood perfusion was measured in rabbit and ischemic hindlimbs for up to 22 days after injection of conditioned PBMC, showing a constant improvement [26]. Furthermore, mouse models of ischemic limbs were implanted with G-CSF-mobilized human PBMC to assess neovascularization by measuring perfusion or capillary density. Both parameters were evaluated for a maximum of 27 days post-injection, showing the greatest improvement during the first week, with a possible stabilization after 3 weeks [62,63]. Our aim was thus to evaluate the stability of cryopreserved PBMCs for a maximum of 6 months, sufficient for multiple thawing and reinfusion, as previously reported [64,65]. In our hands, nucleated cell recovery and monocyte recovery percentages resulted above 80% at all time points, whereas granulocytes decreased by more than 50%, already after one month. The viability of cells always showed remarkably high values, above 99% on average and never below 98% in single measurements. We, therefore, validated long-term storage of frozen PBMCs in nitrogen vapors for up to 1 year.

Altogether, results suggest that the whole manufacturing process can be performed for clinical application. Peripheral blood collected by a physician should be transported at room temperature, and PBMCs should be isolated in a clean room within 8 h of venipuncture. Frozen cells can be stored in nitrogen vapors and thawed for up to 12 months. PBMCs resuspended in 5% human albumin solution should be stored and transported at 4 °C before injection in patients within 24 h to thawing. Hypoxic conditioning of PBMCs should be implemented for clinical application, as it showed a significant enhancement of PBMC functional activity, in particular with increased adhesion, migration, and oxidative stress resistance.

## 4. Materials and Methods

### 4.1. Pre-Clinical Validation

#### 4.1.1. Patient Selection and Informed Consent

Subjects were selected at the Lugano public Regional Hospital. The study involved healthy people undergoing regular blood sampling for routine analysis, as well as patients with cardiovascular diseases willing to donate blood for research purposes. The age of subjects ranged from 23 to 85 years, and 9 were male and 10 female. Written informed consent was obtained from all participants to take part in the study, reviewed by the regional Ethical Committee of the Canton Ticino.

#### 4.1.2. Blood Collection

Peripheral blood was collected by venipuncture in Vacutainer^®^ tubes (Beckton Dickinson, Switzerland) with Acid-Citrate-Dextrose Solution A (ACD-A) as anticoagulant. A total of 40 to 60 mL of peripheral blood was collected by the doctor, packed in an insulated box with temperature stabilizers, and kept at room temperature. Blood samples were transported by a dedicated express carrier (Swissconnect AG, Luzern, Switzerland). All samples were processed within 8 h of venipuncture.

#### 4.1.3. PBCM Isolation by Density Gradient Centrifugation

All reagents and materials used in this experimental approach were selected due to their quality and their upgradeability to GMP conditions. Particular attention was paid to the GMP quality of these products, keeping in mind that the final protocol resulting from these experiments would have to be upgraded for clinical application. The protocol of PBMC isolation was based on separation by density gradient centrifugation with Ficoll^®^, a highly branched polysaccharide. Peripheral blood was diluted 1:1 with Dulbecco’s Phosphate Buffered Saline (DPBS, Merck, Darmstadt, Germany) in 50 mL centrifuge tubes (Falcon^®^, Corning, Switzerland). In parallel, 50 mL tubes were filled with 20 mL of Ficoll^®^-Paque PREMIUM 1.073 (GE Healthcare/Cytiva, Switzerland), and 25 mL of diluted blood was carefully layered on top of the Ficoll paying attention not to mix the two phases. Tubes were then centrifuged at 800 RCF for 30 min with brake off. Phase separation was clearly visible after centrifugation, with platelets and mononuclear cells on top. Plasma on top of interphase was discarded, and the middle layer of buffy coat, containing PBMC, was carefully collected in new tubes. Erythrocytes and granulocytes, having a higher density compared to Ficoll^®^, were left at the bottom of the tube and discarded. PBMC tubes were filled with DPBS with 1% Human Serum Albumin solution (HSA, CSL Behring AG, Switzerland) and centrifuged at 800 RCF for 5 min. Supernatant was then discarded, and two same additional washing steps were performed. Cells were finally resuspended in 5% HSA solution for cell counting and viability measurement.

#### 4.1.4. Cell Counting and Viability Measurement

PBMCs were quantified with an automated device based on propidium iodide (PI) staining and exclusion, the NucleoCounter NC-100 (ChemoMetec, Denmark). For total cell count, PBMCs were lysed and stabilized with reagent A and reagent B (ChemoMetec, Denmark), respectively, to expose all nuclei to PI staining. Counting of dead cells was performed without lysis treatment, and viability was calculated as the proportion between living cells and the total cell count and was expressed as a percentage.

#### 4.1.5. Phenotypic Characterization of PBMC by Flow Cytometry

Characterization of isolated PBMCs was performed by immunophenotyping with a 10-channel flow cytometer Navios (Beckman Coulter, Switzerland). Briefly, 5 × 10^5^ cells previously isolated were centrifuged at 800 RCF for 5 min. The supernatant was discarded, and pellet resuspended in 220 µL of flow cytometry buffer composed of DPBS with 1% human serum off-the-clot (Brunschwig, Switzerland). The following antibodies were incubated with 100 µL of cell suspension for 20 min at room temperature in the dark: CD14-FITC (monocytes), CD56-PE (NK lymphocytes), CD41-APC (Platelets), CD19-APC-A700 (B lymphocytes), CD66b-APC-A750 (granulocytes) and CD3-PB (T lymphocytes). Matched isotypic controls, with no specificity to the target, were used as negative controls. After incubation with the antibody mix, 400 µL of flow cytometry buffer was used to dilute the sample. Finally, to calculate the absolute cell number, 100 µL of PerfectCount^TM^ Microspheres (Cytognos S.L., Salamanca, Spain) were added prior to measurement. The proportion between stained cells and microspheres was used to determine the absolute cell count. Compensation of signal overlap among channels was performed with the LUCID compensation kit (Beckman Coulter, Nyon, Switzerland), applying the negative-positive method with Kaluza software (Beckman Coulter, Switzerland).

#### 4.1.6. PBMC Cryopreservation

PBMCs were cryopreserved in liquid nitrogen gas phase after resuspension in cold (4 °C) freezing medium with CryoSure-DEX40 (WAK-Chemie, Steinbach, Germany) as cryoprotectant. CryoSure-DEX40 contains 55% w/v of Dimethyl Sulfoxide (DMSO) and 5% *w*/*v* of Dextran 40. Cryomedium was composed DPBS with 1% HSA, 11% DMSO, and 1% Dextran 40, prepared with 600 µL of DPBS, 200 µL of 5% human albumin solution, and 200 µL of CryoSure-DEX40 for each cryovial. The desired number of cryogenic tubes and cells was calculated in order to store PBMC with a maximum density of 1 × 10^7^ cells/mL. PBMCs were centrifuged at 800 RCF for 5 min, then the supernatant was discarded, and the pellet was resuspended in freezing medium. One mL of cell suspension was transferred to each Nunc^TM^ cryogenic tube (Thermo Fisher Scientific, Reinach, Switzerland), and PBMCs were immediately frozen in a controlled-rate biofreezer IceCube 14S (Sy-Lab, Purkersdorf, Austria) with the freezing program described in Table 2. Cryovials were subsequently transferred to a cryogenic container and cryopreserved at ≤−150 °C in gas phase of liquid nitrogen.

#### 4.1.7. PBMC Thawing

Cryovials with frozen PBMCs were removed from the cryogenic container and immediately transferred to a 37 °C thermo-block for thawing. For each cryovial, one tube was filled with pre-warmed 5% HSA solution. Approximately one minute later, when a small ice cube was still visible in the tube, washing solution was slowly added and mixed into the cell suspension. Afterward, the entire content of cryovials was transferred to the tubes and centrifuged at 800 RCF for 5 min. A second wash was then performed by addition of washing solution and centrifugation again at 800 RCF for 5 min. PBMCs were finally resuspended in 5% HSA solution for cell counting and viability measurement. Cell recovery percentage was calculated as the proportion between nucleated cells after thawing and the total number of cryopreserved cells. Similarly, monocyte and granulocyte recovery percentages were calculated after quantification by flow cytometry.

#### 4.1.8. Cell Culture

PBMCs were cultured in an incubator at 37 °C with 5% CO_2_ for 24 h. Culture medium was composed of Roswell Park Memorial Institute (RPMI)-1640 medium (Merck, Germany) supplemented with Primocin^TM^ (InvivoGen, Toulouse, France), 10% PLTGold^®^ Human Platelet Lysate (hPL, Merck, Germany) and 2 mM Alanyl-glutamine (Merck, Germany). Cells were detached with TryPle^TM^ Select (Gibco^TM^, Thermo Fisher Scientific, Switzerland) and collected for cell count or further use.

#### 4.1.9. Hypoxic Conditioning

Hypoxic conditioning is the exposure of cells to a low oxygen concentration. This was performed by exposing cells to a gas mixture containing 2% O_2_, 5% CO_2_, and 93% N_2_ (PanGas AG, Dagmersellen, Switzerland). The conditioning system was designed according to previous studies [66,67] in order to maintain a hypoxic environment with cost-effective material. Plastic bags of 30 × 30 cm were used as hypoxic chambers (Solis, Mendrisio, Switzerland). Multi-well plates with cells were placed in the bag, together with a Petri dish filled with distilled water to maintain a humidified atmosphere. A heat sealer PFS 400P (PrimeMatik, Shangai, China), was used to seal the bags. Two holes were drilled at the opposite edges of the bag, one to inflate with the gas mixture and the second to allow air to flow out. The hypoxic chamber was gassed at a pressure of 8–10 psi for 10 min to ensure a complete filling of the bag with the low-oxygen gas mixture. Both holes were finally sealed with the heat sealer, and the hypoxic chambers were incubated for 24 h at 37 °C. The effectiveness of the hypoxic system was tested with an Oakton^TM^ Dissolved Oxygen Meter DO6+ (Cole-Parmer, Wertheim, Germany) by measuring oxygen concentration in culture medium at different time points for up to 24 h.

#### 4.1.10. Adhesion Potential Assay

Adhesion potential of PBMCs was assessed in 48-well plates coated with human fibronectin (Corning, Root, Switzerland). Plates were coated one day before the experiment with 2.5 µg/mL of fibronectin resuspended in RPMI-1640 medium, corresponding to 2.6 µg/cm^2^. Coated plates were left to air-dry at room temperature until next day. PBMC cultured in normoxic and hypoxic conditions were resuspended at a concentration of 1 × 10^6^/mL in RPMI-1640 with 5% hPL. Three to five replicates of 500 µL (5 × 10^5^ cells) were plated on fibronectin-coated wells and cultured at 37 °C in normoxic conditions. Four hours later, cells were washed 3 times with DPBS, and non-adherent cells were collected for cell count. Adherent cells were fixed with 4% paraformaldehyde solution (Electron Microscopy Sciences, Hatfield, PA, USA) for 15 min at room temperature. Excess paraformaldehyde was removed, and cells were washed five times with deionized water. Crystal Violet solution (Merck, Germany) was diluted to 0.5% *w*/*v* in deionized water. Fixed cells were stained for 20 min at room temperature on a bench rocker under slow agitation. Excess solution was removed, cells were washed five times with deionized water, and the plate was air-dried for 24 h. Quantification of adherent cells was performed by colorimetric assay with a microplate reader. Stained cells were incubated with methanol (Merck, Germany) at room temperature for 20 min under constant agitation. Optical density was then measured at 570 nm (OD570) with SpectraFluor Plus microplate reader (Tecan Group AG, Männedorf, Switzerland) and XFluor 4 software (Tecan Group AG, Switzerland).

#### 4.1.11. Migration Potential Assay

Migration potential of conditioned cells was assessed with 6.5 mm polycarbonate cell culture inserts (Nunc^TM^, Thermo Fisher Scientific, Switzerland) with 8 µm pores. After conditioning, PBMCs were resuspended at a density of 1 × 10^6^/mL in serum-free RPMI-1640 medium. The lower chamber was filled with RPMI-1640 medium with 5% hPL and 50 ng/mL of MCP-1 protein (Peprotech, London, UK). The upper chamber was loaded with 300 µL of cell suspension (3 × 10^5^ cells), then plates were incubated at 37 °C with 5% CO_2_ for 2 h. Cell collection was performed in both chambers, and adherent cells were detached with Tryple^TM^. Migrated and non-migrated PBMCs were quantified with NucleoCounter NC-100.

#### 4.1.12. Oxidative Stress Resistance Assay

Oxidative stress resistance of normoxic and hypoxic PBMCs was assessed by incubation with a potent generator of oxidative stress, such as hydrogen peroxide (H_2_O_2_). Conditioned cells were resuspended at a density of 1 × 10^6^/mL in RPMI-1640 with 5% hPL, and three replicates of 200 µL (2 × 10^5^ cells) were seeded in 96-well plates. A hydrogen peroxide solution (Merck, Germany) was added to obtain a final concentration of 400 µM. Cells were then incubated for 24 h in normoxic conditions and subsequently detached with Tryple^TM^. Cell number and viability were measured with NucleoCounter NC-100.

#### 4.1.13. Measurement of Reactive Oxygen Species (ROS)

ROS induced by hypoxic conditioning and oxidative stress were measured by incorporation of 2′,7′-Dichlorofluorescin diacetate (DCF-DA, Merck, Germany). PBMC cultured in normoxic and hypoxic conditions were resuspended at a density of 1 × 10^6^/mL in RPMI-1640, and three to five replicates of 200 µL (2 × 10^5^ cells) were seeded in 96-well plates. DCF-DA was diluted and added to obtain a final concentration of 20 µM in each well. Plates were incubated on a shaker in the dark at 37 °C. After 30 min, cells were washed three times with DPBS and resuspended in 200 µL of culture medium. Excitation was performed at 485 nm, and emission was measured at 525 nm with SpectraFluor Plus microplate reader and XFluor 4 software. Subsequently, ROS was measured after oxidative stress resistance assay using the same protocol. Negative controls were performed with unstained cells to assess autofluorescence, as well as with cell-free combinations of growth medium, DPBS and hydrogen peroxide stained with DCF-DA. Results are expressed as relative fluorescence, normalized to normoxic cultures after conditioning.

#### 4.1.14. RNA Extraction and Reverse Transcription

Total RNA was extracted from PBMC with RNeasy Mini Kit (Qiagen, Hilden, Germany) following manufacturer’s instructions. RNA concentration was quantified by spectrophotometric analysis with a Genova MK3 spectrophotometer (Jenway^TM^, Cole-Parmer, Germany). RNA was diluted 10 times in RNase-free water before measurement of absorption at 260 nm, as well as 230 and 280 nm for purity assessment. RNA concentration was calculated through Absorbance = C × l × ε, where C is the concentration of nucleic acid, l is the path length of the cuvette (1 cm), and ε is the extinction coefficient of RNA corresponding to 0.025 (mg/mL)^−1^cm^−1^.

Reverse transcription of extracted RNA was performed using Maxima H Minus First Strand cDNA Synthesis Kit (Thermo Fisher Scientific, Switzerland), according to manufacturer’s instruction, in a total volume of 20 µL. Samples were used immediately for qPCR analysis or stored at −20 °C for further use. Two negative controls were performed with all components of the reverse transcription by exclusion of Maxima enzyme mix and RNA template.

#### 4.1.15. Reverse Transcription Polymerase Chain Reaction (RT-PCR)

PCR reactions were carried out with SsoAdvanced SYBR Green Supermix (Bio-Rad Laboratories, Fribourg, Switzerland). ß-actin was used as housekeeping gene (Mycrosynth AG, Balgach, Switzerland) and the following primers (Bio-Rad Laboratories) targeting human genes were used: *VEGF-A* (qHsaCED0043454), *FGF-2* (qHsaCED0056993), *PDGF-A* (qHsaCID0005879), *CCR2* (qHsaCED0043920), *CXCR4* (qHsaCED0002020), *TNF-ɑ* (qHsaCED0037461), *IFNG* (qHsaCID0017614), *IL-1b* (qHsaCID0022272) and *IL-2* (qHsaCID0015409).

Two duplicates of 10 µL were transferred to a 96-well plate (Hard-Shell Thin-Wall 96-well Skirted PCR Plates, Bio-Rad Laboratories, Switzerland) and sealed with a plastic microplate sealer (Bio-Rad Laboratories, Switzerland). Plates were then centrifuged at 1000 RCF for 1 min and processed with a CFX Connect Real Time PCR Detection System (Bio-Rad Laboratories, Switzerland). Cycling conditions were set according to primer’s specifications as follows: 95 °C for 2 min followed by 40 cycles consisting of 10 s at 95 °C and 30 s at 60 °C. Melt-curve was generated by raising temperature at 0.5 °C/5 s from 65 °C to 95 °C. CFX Connect Software (Bio-Rad Laboratories, Switzerland) was used for analysis, and the ΔΔCT method was applied. Raw threshold cycles (CT) were normalized to the housekeeping gene (β-actin), then ΔCT was calibrated to control samples to obtain ΔΔCT values.

#### 4.1.16. Statistical Analysis

Results are presented as mean ±SEM (standard error of the mean) from triplicate determinations. An unpaired two-tails Student’s *t*-test was used to compare the average of two groups. One-way ANOVA followed by Tukey HSD post hoc test was used for statistical analysis of multiple groups and to assess the reproducibility among replicates. Two-way ANOVA followed by Tukey HSD post hoc test was applied in case of multiple factors. A *p*-value < 0.05 was considered statistically significant, * *p* < 0.05, ** *p* < 0.01, *** *p* < 0.001. Statistical analysis was carried out with Prism GraphPad software (GraphPad, San Diego, CA, USA) and with Real Statistics Resource Pack (Microsoft, Redmond, WA, USA).

### 4.2. GMP-Compliant Validation

Validation of relevant processes is required for the authorization of production by the regulatory body, in our case, the Swiss Agency for Therapeutic Products. Validations require a documented plan to describe the experiment and illustrate all the materials, facilities, and equipment used, as well as a validation report summarizing the results. The experiments require at least 3 different and independent samples or measurements, as mentioned in Annex 15 to the EU/PIC/S Guide. We evaluated relevant processes considering the whole procedure, beginning with blood collection, transport, cell processing, cryopreservation, thawing, and cell stability before injection and in liquid nitrogen. Validation experiments were performed in an authorized clean room (Technopark, Zurich, Switzerland) under GMP-compliant quality assurance system and covered all the following processes: pre-processing time and temperature, PBMC isolation, cryopreservation and thawing, PBMC storage before injection and long-term storage of frozen PBMC.

#### 4.2.1. Validation of Pre-Processing Time and Temperature

Peripheral blood drawn from 3 different patients was tested for an 8 h delayed processing. PBMC isolation was performed twice, once within one hour of venipuncture and the second after 8 h of storage at room temperature (23 ± 5 °C). The evaluation was based on two different methods, the Nucleocounter measured the total number of nucleated cells and viability in the PBMC solution, and a flow cytometric analysis measured monocyte and granulocyte number. Results of freshly isolated PBMCs were compared with those obtained after 8 h.

#### 4.2.2. Validation of PBMC Isolation, Cryopreservation, and Thawing

PBMC isolation, cryopreservation, and thawing of 3 different and independent samples were performed in an aseptic environment, laminar flow cabinet EU GMP grade A in an EU GMP clean room grade B. Processing was performed according to SOP of a GMP-compliant quality management system (Swissmedic authorization nr. 510270). Total number of nucleated cells and viability in the PBMC solution were measured with Nucleocounter, and monocyte and granulocyte numbers were assessed by flow cytometry immunophenotyping, and results compared with those obtained in pre-clinical research.

Three different and independent samples processed in clean room under normal operating conditions were cryopreserved in vapors of nitrogen at ≤−150 °C for 4 to 13 weeks and then thawed and characterized by measuring recovery percentage of PBMC, viability of recovered cells, as well as recovery percentages of monocytes and granulocytes. Results were compared with those obtained before freezing and cryopreservation, as well as results obtained in the pre-clinical research.

#### 4.2.3. Validation of Storage Time for Thawed PBMC before Injection

Storage of thawed PBMCs was tested at 4 °C for 24 h to determine the maximum transport time before injection and a possible effect on the product quality. Cells were resuspended in 5% human albumin solution for injection and kept in syringes at 4 °C. Cell count, viability, and immunophenotyping of PBMCs were performed immediately after thawing and after a 24 h storage for comparison.

#### 4.2.4. Validation of Long-Term Storage of Cryopreserved PBMC

Cryopreservation of the product was evaluated for up to one year to determine the effect of cryopreservation time on the quality of the sample at 1, 2, 3, 6, and 12 months. The evaluation was based on average viability and recovery of nucleated cells, monocytes, and granulocytes, expressed as relative count normalized to cell numbers before cryopreservation.

## Figures and Tables

**Figure 1 ijms-23-12669-f001:**
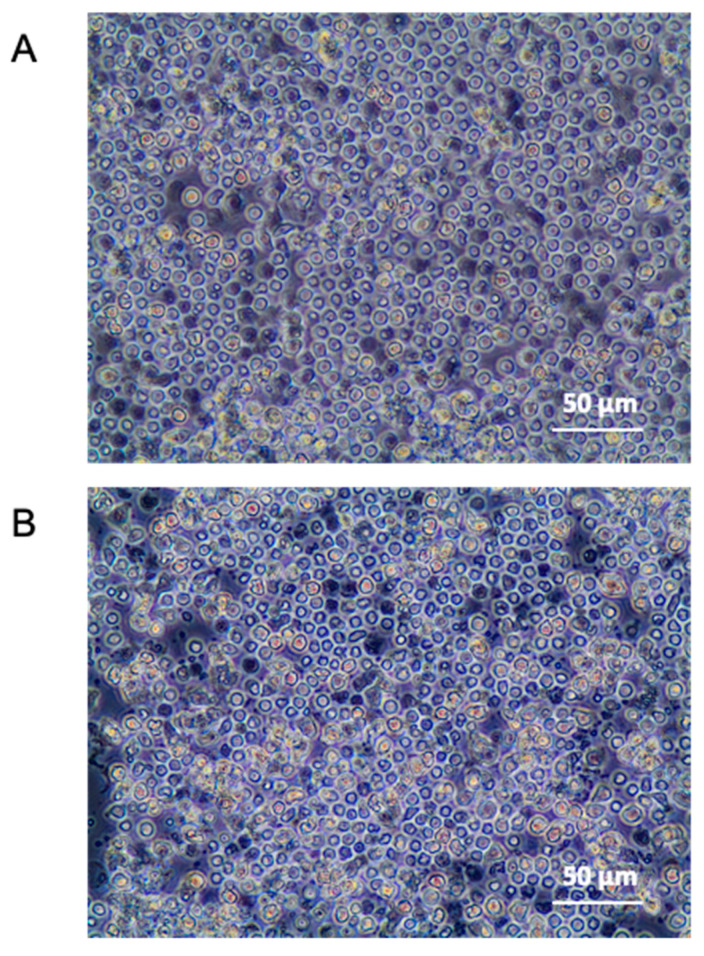
Representative morphology of PBMC cultured for 24 h in (**A**) normoxic (5% CO_2_, 95% air) and (**B**) hypoxic (5% CO_2_, 2% O_2_, 93% N_2_) conditions. Magnification 200×.

**Figure 2 ijms-23-12669-f002:**
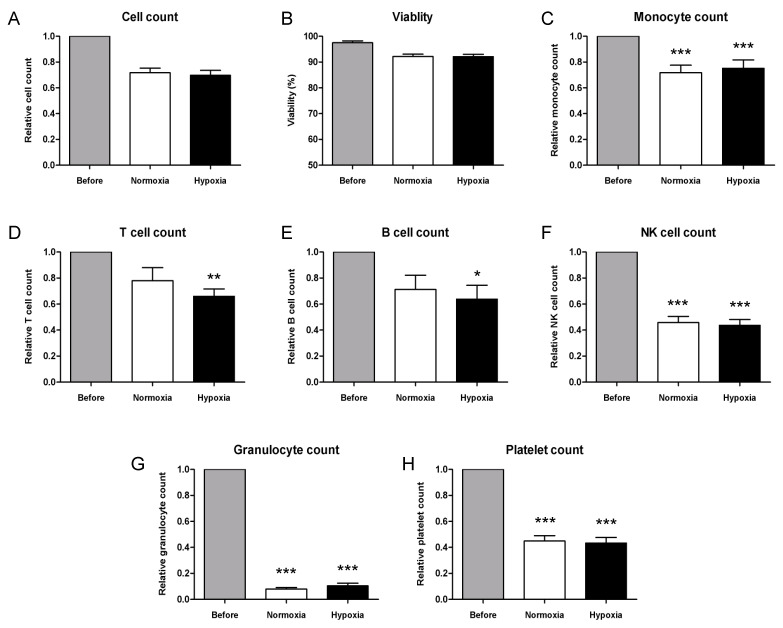
Immunophenotyping and quantification of PBMC before conditioning and after a 24 h culture in normoxic (5% CO_2_, 95% air) and hypoxic (5% CO_2_, 2% O_2_, 93% N_2_) environments. (**A**) Relative cell count, (**B**) viability percentage and relative count of (**C**) monocytes, (**D**) T cells, (**E**) B cells, (**F**) NK cells, (**G**) granulocytes and (**H**) platelets. Relative cell count is normalized to PBMC number before conditioning. Data are represented as mean ±SEM on N = 15 samples. Statistically significant differences were observed between normoxia or hypoxia compared to control (Before). * = *p* < 0.05, ** = *p* < 0.01, *** = *p* < 0.001.

**Figure 3 ijms-23-12669-f003:**
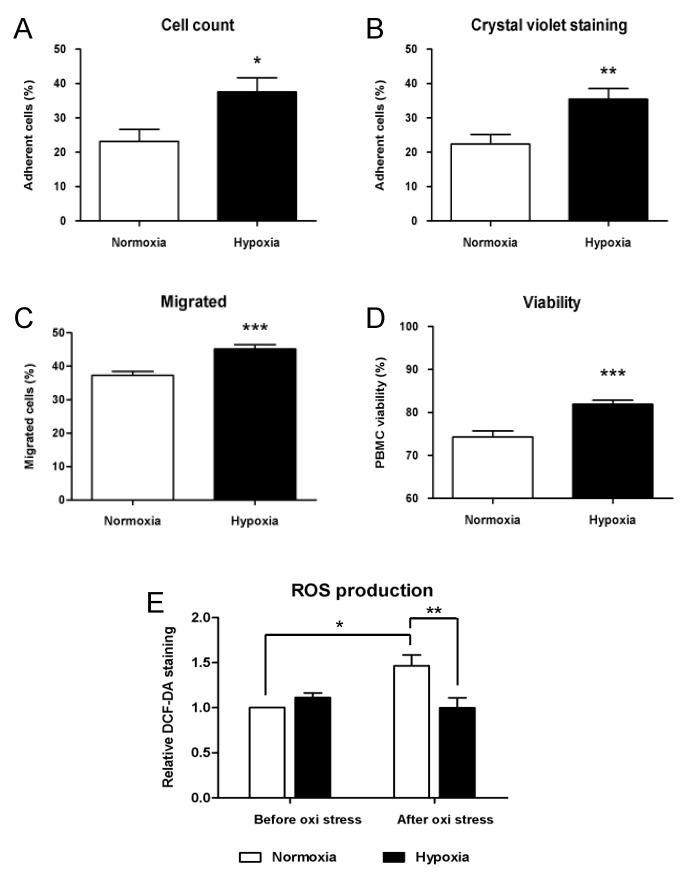
Functional analysis of PBMC conditioned in normoxic (control) and hypoxic environments. (**A**) Calculation of adhesion percentage by subtraction of non-adherent cells in N = 7 samples. (**B**) Measurement of adhesion percentage by crystal violet staining and colorimetric assay in N = 7 samples. (**C**) Average percentage of migrated PBMC in N = 8 samples. (**D**) Viability of PBMC before and after a 24 h incubation with 400 µM H_2_O_2_ in N = 14 samples. (**E**) Relative quantification of ROS production before and after a 24 h incubation with 400 µM H_2_O_2_ in N = 8 samples. DCF-DA fluorescent emission was used for quantification. Data are represented as mean ± SEM. * = *p* < 0.05, ** = *p* < 0.01, *** = *p* < 0.001.

**Figure 4 ijms-23-12669-f004:**
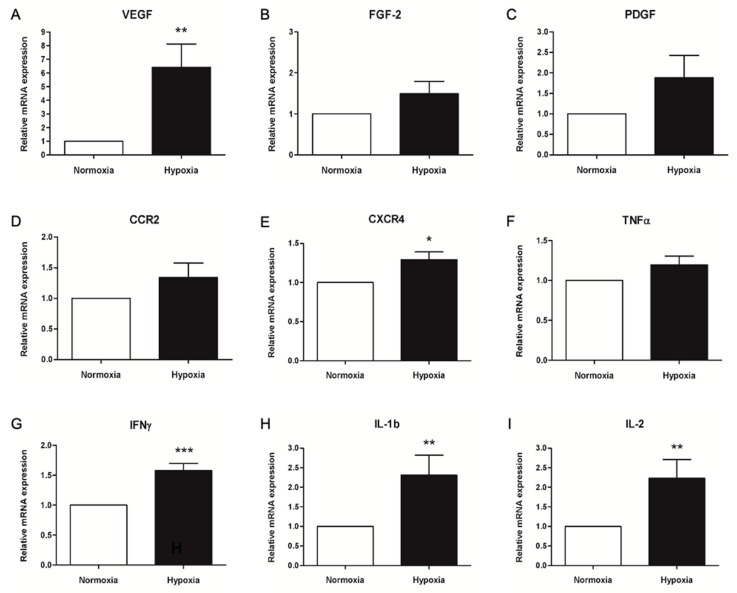
Relative quantification of mRNA expression in normoxic (control) and hypoxic cultures obtained by quantitative RT-PCR. β-actin was used as housekeeping gene. (**A**) Relative expression of (**A**) VEGF-A, (**B**) FGF-2, (**C**) PDGF, (**D**) CCR2, (**E**) CXCR4, (**F**) TNF-α, (**G**) IFN-γ, (**H**) IL-1β and (**I**) IL-2 genes. Values are normalized to normoxic controls with the ⊗⊗CT method. Data are represented as mean ± SEM from triplicate determinations, N = 8. * = *p* < 0.05, ** = *p* < 0.01, *** = *p* < 0.001.

**Figure 5 ijms-23-12669-f005:**
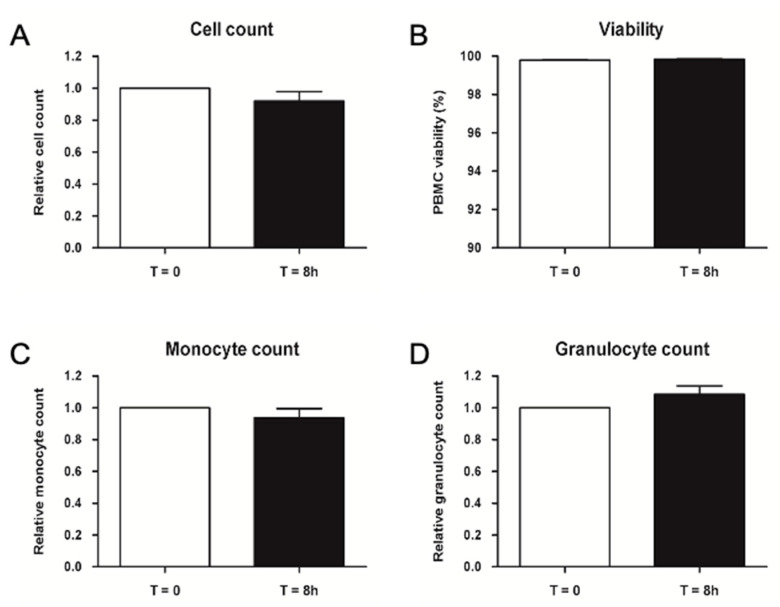
Validation of the 8 h storage at room temperature of peripheral blood before processing. Average cell count (**A**), viability (**B**), monocyte (**C**) and granulocyte (**D**) count of PBMC isolated at T = 0 and T = 8 h. Relative values are normalized to control samples isolated at T = 0. Data are represented as mean ± SEM, N = 3.

**Figure 6 ijms-23-12669-f006:**
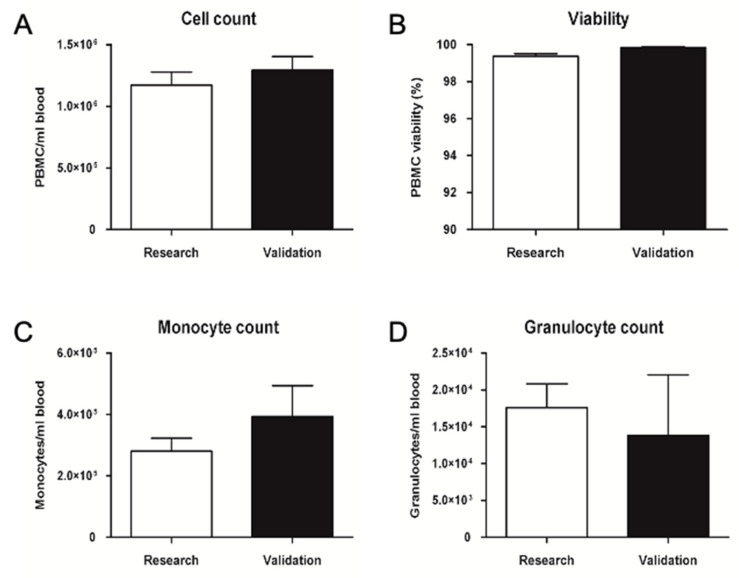
Validation of PBMC isolation in a GMP-compliant aseptic environment (clean room). Average cell count (**A**), viability (**B**), monocyte (**C**) and granulocyte (**D**) count of PBMC isolated from peripheral blood. Data are represented as mean ± SEM. Research N = 11, Validation N = 3.

**Figure 7 ijms-23-12669-f007:**
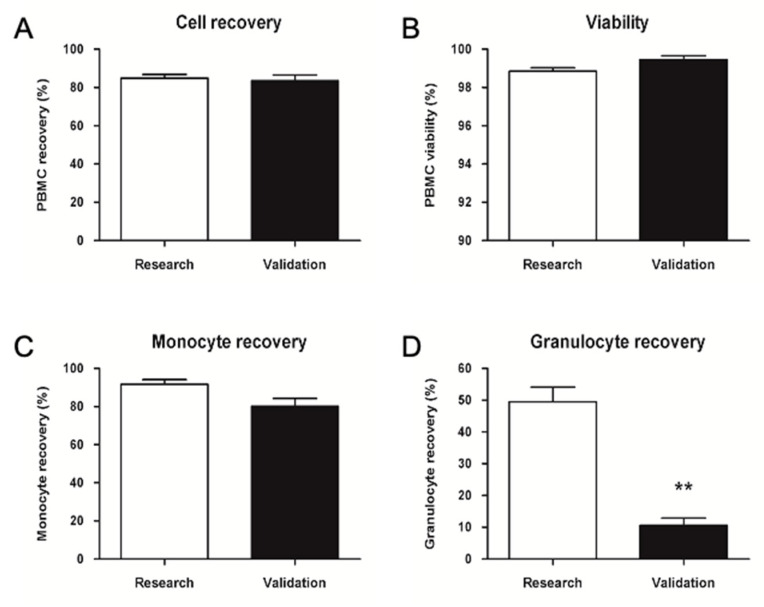
Validation of PBMC cryopreservation and thawing in a GMP-compliant aseptic environment (clean room). Average recovery percentage of nucleated cells (**A**), viability (**B**), monocyte (**C**) and granulocyte (**D**) recovery percentages. Date are represented as mean ± SEM. Research N = 27, Validation N = 3. ** = *p* < 0.01.

**Figure 8 ijms-23-12669-f008:**
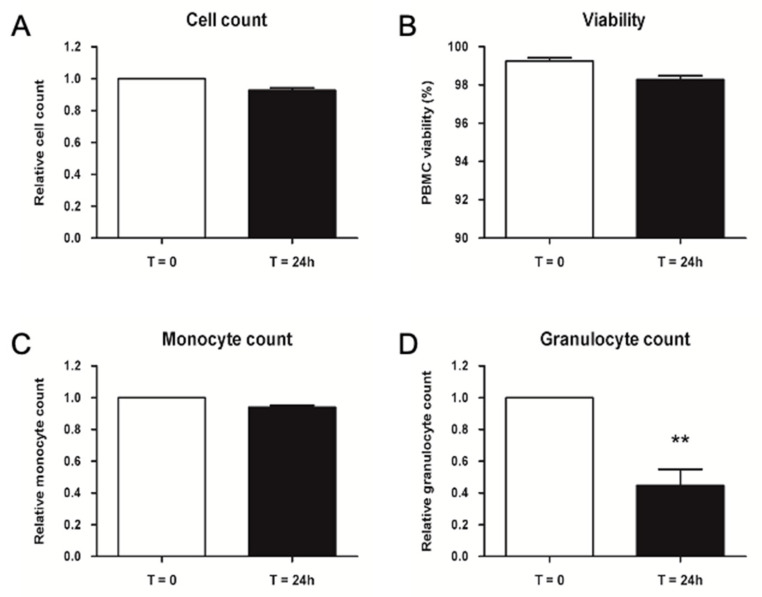
Validation of PBMC storage at 4 ± 2 °C for 24 h. (**A**) Relative cell count, (**B**) viability percentage, (**C**) relative monocyte and (**D**) granulocyte count of thawed PBMC at T = 0 and T = 24 h. Relative values are normalized to control samples at T = 0. Data are represented ad mean ± SEM, N = 3. ** = *p* < 0.01.

**Figure 9 ijms-23-12669-f009:**
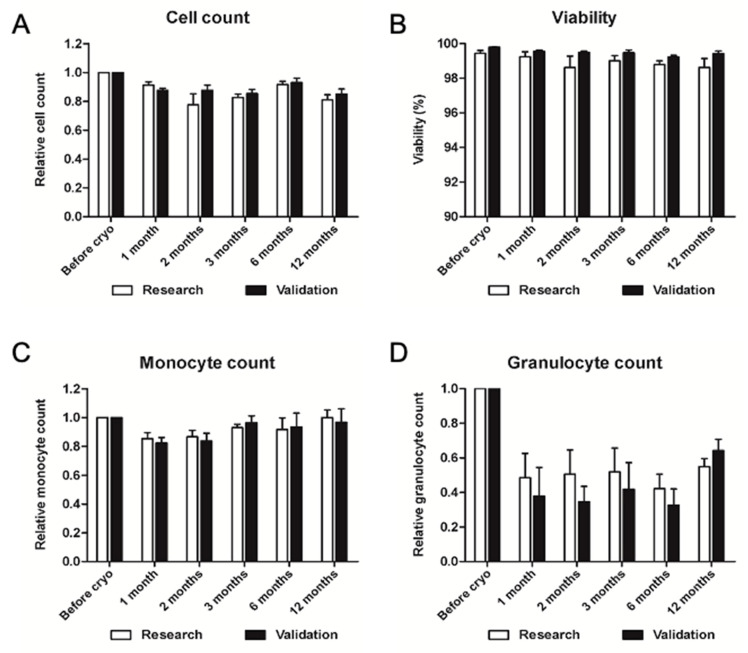
Validation of long-term storage of PBMC cryopreserved in nitrogen vapors (≤−150 °C). Relative cell count (**A**), viability percentage (**B**), relative monocyte (**C**) and granulocyte count (**D**) of PBMC before cryopreservation and at T = 1, 2, 3, 6, 12 months. Relative values are normalized to control samples before freezing. Date are represented ad mean ±SEM, N = 3.

**Table 1 ijms-23-12669-t001:** Average proportion and quantification of Ficoll-purified PBMC populations by flow cytometry immunophenotyping before and after cryopreservation. N = 15.

	Before Freezing		After Thawing	
	%	Cells/mL Blood	%	Cells/mL Blood
Monocytes	11.36	2.67 × 10^5^	17.25	2.72 × 10^5^
T lymphocytes	37.79	6.69 × 10^5^	35.04	4.17 × 10^5^
B lymphocytes	10.69	1.73 × 10^5^	12.59	1.63 × 10^5^
NK lymphocytes	10.29	1.93 × 10^5^	15.86	1.98 × 10^5^
Platelets	24.27	6.03 × 10^5^	23.42	3.99 × 10^5^
Granulocytes	0.47	1.10 × 10^4^	0.45	7.23 × 10^3^

**Table 2 ijms-23-12669-t002:** PBMC freezing program of the controlled rate biofreezer IceCube 14 S.

STEP	Temperature (°C)	Time (min)	Heat	Hold
1	4.0	2	Off	Stop
2	4.0	5	Off	
3	−6.0	10	Off	
4	−45.0	1.5	Off	
5	−20.0	2.5	On	
6	−45.0	25	Off	
7	−120.0	5.5	Off	
8	−120.0	5	Off	

## Data Availability

The study did not report any patient’s data.

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
