# Peer review of "Upgrading Monocytes Therapy for Critical Limb Ischemia Patient Treatment: Pre-Clinical and GMP-Validation Aspects"

_ijms, 2022, doi:10.3390/ijms232012669_

Round 1

Reviewer 1 Report

The authors report upgrading monocytes therapy for clinical limb ischemia patient treatment.

1.      This study was lacked ethics committee/institutional review board approved numbers. Please confirm these things.

2.      The research design was not clear in this study. The authors should show the outline of this study in a Figure.

3.      In the Figure 3, the authors should provide the representative immunohistochemical data of crystal violet staining and ros production.

4.      In the Figure, the authors should provide the data of gating strategies by frow cytometry. Also, the authors should provide the data of quadrant gates.

5.      The authors should provide the ELISA data in Figure 4.

6.      The authors should analyze colony counting and provide the data in Figure 9.

Reviewer 2 Report

An interesting paper on a very popular topic. However, let me make a few remarks.

“An important advantage of PBMC is the non-invasive approach of blood collection...”

Do you really mean that venipuncture is non-invasive?

“The study involved healthy people... The age of subjects ranged from 23 to 85 years...”

It is appropriate to mention the gender of patients.

“A low-cost home-made system for hypoxic conditioning was designed and tested”

It is remarkable. But for manuscript claiming the results of preclinical studies and their adaptation to GMP, this sounds unconvincing.

Fig. 2, 3. “p < 0.05, ... p < 0.01” - difference between normoxia and hypoxia? Or “Before?” This should be clear in all cases.

Figures 3, 5, 6, 7, 8 - too, disproportionately large.

There are a number of sloppiness and unsuccessful phrases in the text that reduce the overall impression. Here is some of them.

Fig.4. “... Values are normalized to normoxic controls with the CT method”. Please correct.

“Figure 7. an average cell recovery above 80% and similar to the average calculated on 27 thawing processes of 7 different research samples. Average viability difference was below 1%. Monocyte recovery was above 80% and slightly lower than the average obtained in the research process. Granulocyte recovery percentage was significatively lower compared to research.”

Do you really think that this is a caption for Figure? 

Round 2

Reviewer 1 Report

The authors report upgrading monocytes therapy for clinical limb ischemia patient treatment.

1.      This study was lacked ethics committee/institutional review board approved numbers. Please confirm these things.

2.      The research design was not clear in this study. The authors should show the outline of this study in a Figure.

3.      In the Figure 3, the authors should provide the representative immunohistochemical data of crystal violet staining and ros production.

4.      In the Figure, the authors should provide the data of gating strategies by frow cytometry. Also, the authors should provide the data of quadrant gates.

5.      The authors should provide the ELISA data in Figure 4.

6.      The authors should analyze colony counting and provide the data in Figure 9.

Author Response

It seems that the observations of Reviewer 1 in round 2 are the same as in round 1. Included the same answers as round 1. Please let us know wether something went wrong.
